# Effects of Slow-Acting Metformin Treatment on the Hormonal and Morphological Thyroid Profile in Patients with Insulin Resistance

**DOI:** 10.3390/pharmaceutics14101987

**Published:** 2022-09-20

**Authors:** Rosita A. Condorelli, Rossella Cannarella, Antonio Aversa, Livia Basile, Ottavia Avola, Aldo E. Calogero, Sandro La Vignera

**Affiliations:** 1Department of Clinical and Experimental Medicine, University of Catania, 95123 Catania, Italy; 2Department of Experimental and Clinical Medicine, Magna Graecia University of Catanzaro, 88100 Catanzaro, Italy

**Keywords:** metformin treatment, thyroid hormone, thyroid nodules, insulin resistance

## Abstract

Metformin appears to reduce TSH levels in untreated hypothyroid patients. In contrast, in euthyroid patients with type 2 diabetes mellitus (T2DM), metformin is initially devoid of effects on TSH. However, it is followed by a significant reduction in TSH level after twelve months of treatment. Additionally, some evidence suggests that metformin may also improve thyroid morphological abnormalities. This study aimed to evaluate the effects of metformin not only on TSH and thyroid hormone values, but also on thyroid volume and nodules. A total of 50 patients (mean age: 36.9 ± 12.8 years) with insulin resistance (homeostatic model assessment (HOMA) index ≥2.5) and with thyroid uninodular disease were recruited for this study. They were prescribed slow-acting metformin at a daily dose of 500 mg for six months. Treatment with metformin in euthyroid patients with uninodular thyroid disease and insulin resistance reduces TSH levels, increases FT4 and FT3 values, and decreases thyroid and nodule volumes. These data suggest that metformin may be an effective drug not only for the treatment of T2DM and metabolic syndrome, but also for thyroid disease.

## 1. Introduction

Obesity, type 2 diabetes mellitus (T2DM), and thyroid diseases are among the most common endocrine disorders [1], often occurring together in the same patient. In particular, the correlation between diabetes and hypothyroidism has long been recognized. In this regard, several prospective studies stated a remarkably high prevalence (10–15%) of thyroid dysfunction in patients with T2DM [2,3].

Metformin is a biguanide derivative, commonly used as an oral hypoglycemic agent for the treatment of T2DM. In recent decades it has emerged as a drug with multiple properties [4]. Metformin provides an anti-inflammatory and anti-cancer action, improving glycemic and lipid metabolism [5]. Vigersky and colleagues were the first to randomly evaluate a possible link between metformin and TSH serum levels in hypothyroid patients on levothyroxine (LT4) treatment [6]. Since then, various studies have been published, mainly to corroborate or reject its effect on thyroid function [7,8,9,10,11,12,13]. Most of these studies were of prospective or retrospective design and some included only a limited number of patients. 

The effect of metformin was evaluated in both LT4-treated and non-LT4-treated patients with hypothyroidism. Metformin appears to reduce TSH levels in untreated hypothyroid patients. In particular, the administration of metformin to four patients with untreated chronic hypothyroidism led to a decrease in the serum of TSH levels below reference values, without clinical signs of hyperthyroidism or a relevant change in free T4 (FT4) and free T3 (FT3) [6]. Another retrospective study showed that metformin administration resulted in a statistically significant decrease in TSH, together with an increase in FT4 [14]. The authors of this study suggested that the suppression of TSH levels appeared to be independent of the increase in FT4. According to this study, the short-term administration of metformin in women with diabetes, obesity and primary hypothyroidism who did not receive LT4 was associated with a significant drop in TSH that disappeared when metformin was discontinued, whereas thyroid hormone (TH) levels did not change.

Some studies found that metformin administration did not affect TSH in euthyroid patients with T2DM [14] and on subclinical hypothyroidism [15], while it reduces TSH in hypothyroid patients. However, a meta-analysis [16] did not confirm a relationship between TSH values and metformin treatment in euthyroid patients. Consequently, this discrepancy can be explained by what the authors called the “buffer effect” of metformin, since metformin induces a return of circulating TSH at the middle of the normal range and indicates 2.98 mU/L as the threshold level. That is, patients with a serum TSH level above 2.98 mU/L will experience a metformin-reducing effect on TSH, whereas individuals with basal TSH levels below 2.98 mU/L will experience an opposite effect [17]. In a double-blind placebo-controlled clinical trial in 89 patients with prediabetes, the administration of metformin for three months was associated with a decrease in serum TSH levels only in those patients with a TSH value higher than 2.5 μU/mL [18]. Furthermore, this TSH-lowering effect of metformin with coexisting T2DM was not observed in subclinical hyperthyroid patients [19]. The data in the literature are still very contradictory and, therefore, further supporting evidence is needed.

Some evidence suggests that metformin treatment may also improve thyroid morphological abnormalities [20], even if the data in the scientific literature on the possible effect of metformin on thyroid volume and nodules are still few, especially in the euthyroid population. Therefore, this study was undertaken to evaluate the effects of metformin not only on TSH and TH levels, but also on thyroid volume and nodules in a Sicilian, euthyroid population with insulin resistance and uninodular thyroid disease. 

## 2. Patients and Methods

### 2.1. Study Protocol

This is a prospective study carried out in patients >18 years who were referred to the Division of Endocrinology, Metabolic Diseases and Nutrition of the University-Teaching Hospital Policlinico “G. Rodolico-San Marco”, University of Catania (Catania, Italy), for weight control.

Each patient underwent an evaluation of the body mass index (BMI), waist circumference (WC), glycemia, insulin, TSH, FT4, and FT3 serum levels. A thyroid ultrasound was also performed to evaluate the presence of thyroid nodules; only patients with insulin resistance (homeostatic model assessment (HOMA) index ≥ 2.5) and thyroid uninodular disease were included. In particular, the thyroid ultrasound was used to measure the volume of the thyroid gland and that of the thyroid nodule. The HOMA index was calculated using the following formula: [glycemia (mg/dL) × insulin (µIU/mL)]/405. 

The patients included in the study were prescribed slow-acting metformin at a daily dose of 500 mg for six months. The slow-release formulation was preferred for a lower prevalence of side effects (particularly diarrhea) that are frequently reported by patients in clinical practice. The patients were also prescribed a Mediterranean diet. The protocol provided consisted of 45–60% of carbohydrates, mainly complexes (such as starches from cereals), 10–12% of proteins, corresponding to 0.9 g per kg of body weight; 20–35% of fats with a percentage of saturated fats (mostly comprising animal products, except for fish) of less than 10%. The number of calories introduced was established according to the calculation of the daily caloric requirement with a consequent reduction in the estimated income by 20%. After six months, each patient was evaluated for BMI, WC, glycemia, and insulin to calculate the HOMA index, TSH, FT4, and FT3 serum levels. Each of them also underwent a thyroid ultrasound scan and the thyroid gland and nodule volumes were calculated.

### 2.2. Patient Selection

The inclusion criteria were the presence of insulin resistance (HOMA index ≥ 2.5), uninodular thyroid disease, euthyroidism (normal TSH and TH values), and age > 18 years. Furthermore, only patients with solid, hypoechoic, and homogeneous nodules were included. 

Patients were excluded in case of comorbidities (e.g., dyslipidemia, hypertension, diabetes mellitus), positivity for thyroid antibodies, and a positive family history of thyroid cancer. Cystic and mixed cystic thyroid nodules, hyperechoic and isoechoic nodules, and those with inhomogeneous structure or irregular contours were excluded. The nodules suspected of malignancy underwent fine-needle aspiration cytology (FNAC), and the patients were only admitted to the study when the cytology showed a TIR 2 result. 

### 2.3. Hormonal Measurements

Blood tests were performed after 12 h of fasting, and the samples were analyzed at the central laboratory of the University-Teaching Hospital Policlinico, University of Catania (Catania, Italy). A hormonal evaluation was performed by electrochemiluminescence (Hitachi-Roche equipment, Cobas 6000, Roche Diagnostics, Indianapolis, IN, USA). The reference values were as follows: TSH 0.4–4.2 µUI/mL, FT4 6.8–16 pmol/L, and FT3 3.8–6 pmol/L. 

### 2.4. Ultrasound Scan

Thyroid volume was calculated using the formula [antero-posterior (AP) diameter of the right lobe × transverse (T) diameter of the right lobe × longitudinal (L) diameter of the right lobe × 0.52] + (AP diameter of the left lobe + T diameter of the left lobe × L diameters of the left lobe × 0.52). The volume of the nodules was evaluated using the formula: AP diameter × T diameter × L diameter × 0.52. The ultrasound used for the thyroid assessment was a Logiq S8, GE, which was equipped with a linear probe with a frequency of 7.5 MHz. 

### 2.5. Statistical Analysis

The results are reported as mean ± SD throughout the study. The normality of the variables was assessed using the Shapiro–Wilks test. The Student’s *t*-test or Wilcoxon test for paired samples was used to evaluate the difference in the before and after values of BMI, WC, HOMA index, TSH, FT4, FT3, thyroid volume, and nodule volume. Subsequently, a multi-regression analysis was undertaken to evaluate whether the difference between the ratio and delta (Δ) of the before and after values of the thyroid and the nodule volumes were significantly associated with the ratio or the Δ of the before and after values of BMI, WC, HOMA index, TSH, FT4, and FT3. The ratio of each variable was calculated using the following formula: [(after value/before value) × 100]. The Δ of each variable was calculated using the formula: (before value—after value). Finally, a *p*-value lower than 0.05 was accepted as statistically significant. The statistical analysis was performed using MedCalc (Medcalc Software Ltd, Version 19.6–64 bit, Ostend, Belgium).

### 2.6. Ethical Approval

This study was conducted at the Division of Endocrinology, Metabolic Diseases and Nutrition of the University-Teaching Hospital Policlinico “G. Rodolico-San Marco”, University of Catania (Catania, Italy). The protocol was approved by the internal Institutional Review Board. Informed consent was obtained from each participant after a full explanation of the purpose and nature of all the procedures used. The study was conducted according to the principles expressed in the Declaration of Helsinki. 

## 3. Results

A total of 50 patients (mean age: 36.9 ± 12.8 years) were recruited for this study. Of these, 10 were male (mean age: 30.5 ± 12.1 years) and 40 were female (mean age: 38.5 ± 12.6). Their baseline characteristics are shown in Table 1. No difference was found between the two genders (Table 1).

### 3.1. Before and after Analysis

Treatment with metformin significantly reduced BMI, WC, and HOMA index (Figure 1, Panels A–C). Interestingly, the treatment resulted in a significant reduction in serum TSH levels and a significant increase in both FT3 and FT4 (Figure 1, Panels D–F). 

The thyroid and nodule volume analyses were consistent with a significant reduction in both after metformin treatment (Figure 2, Panel A and B, respectively). 

### 3.2. Correlation Analysis

A multi-regression analysis was then performed to evaluate whether the reduction in the thyroid or nodule volumes was associated with the reduction in BMI, CV, HOMA index, and TSH, and with the increase in the serum levels of FT4 and FT3. 

When the before and after ratio of thyroid volume was considered as a dependent variable, we found a significant positive association with the ratios of the HOMA index and TSH and a negative correlation with the FT4 ratio (Figure 3 and Table 2). Furthermore, the Δ of thyroid volume was positively associated with Δ WC and Δ TSH, and negatively associated with the serum levels of Δ FT4 (Table 2). 

As for the volume of the nodules, its before and after ratio was positively associated only with the FT3 ratio (Figure 4 and Table 3). However, when the Δ of the nodule volume was considered, it was positively associated with both Δ BMI and Δ FT3 (Table 3). 

The dependent variable was the ratio of thyroid volume (%) when the independent variables were analyzed as ratios. The dependent variable was the difference (Δ) of the thyroid volume when the independent variables were analyzed as Δ. The ratio of each variable was calculated using the following formula: [(after value/before value) × 100]. The difference of each variable was calculated using the formula: before value—after value. Legend: BMI, body mass index; FT3, free-triiodothyronine; FT4, thyroxin; HOMA, homeostatic model assessment; TSH, thyroid-stimulating hormone; WC, waist circumference.

The dependent variable was the ratio (%) of thyroid nodule volume when the independent variables were analyzed as ratios. The dependent variable was the difference (Δ) of the thyroid nodule volume when the independent variables were analyzed as Δ. The ratio of each variable was calculated using the following formula: [(after value/before value) × 100]. The difference of each variable was calculated using the following formula: before value—after value. Legend: BMI, body mass index; FT3, free-triiodothyronine; FT4, thyroxin; HOMA, homeostatic model assessment; TSH, thyroid-stimulating hormone; WC, waist circumference.

## 4. Discussion

The present study showed that treatment with slow-acting metformin in euthyroid patients with uninodular thyroid disease and insulin resistance leads to a reduction in TSH levels and an increase in FT4 and FT3 serum levels. Various studies have been performed with the aim of elucidating the role of metformin treatment on TSH levels [11,21,22]. However, the results of these studies are contradictory due to their methodological discrepancy [16]. Since weight loss is recognized to influence thyroid function tests, it was hypothesized that the metformin-induced reduction in TSH could be based on the weight loss experimented by study participants [23,24,25]. 

Most studies reported that metformin reduced TSH levels in hypothyroid patients and had no effect on euthyroid patients, but this aspect could be influenced by the duration of the treatment. Indeed, a population-based longitudinal study showed that metformin treatment resulted in an increased prevalence of low TSH levels only in patients with hypothyroidism on thyroxine, while no effect was found in euthyroid patients. Interestingly, this reduction in TSH was greater in the first 90–180 days of therapy [26]. These results agree with the findings of another study showing a more important reduction in TSH levels in euthyroid patients after twelve months of metformin administration [27].

In a prospective study by Cappelli and colleagues, 101 patients with T2DM were examined. Before the initiation of metformin treatment, the participants were divided into three groups: the first group included 29 patients with hypothyroidism who received thyroxine; the second consisted of 18 untreated patients with subclinical hypothyroidism; and the third group comprised 54 euthyroid patients. Treatment was prescribed for at least one year. The results showed a significant reduction in TSH levels only in patients with T2DM and hypothyroidism, either treated or untreated. Furthermore, after metformin treatment, no significant variations were found in serum FT4 levels and the BMI did not change significantly, confirming that metformin acts regardless of the weight loss it may induce [8]. In the same line, a cross-sectional study by Díez & Iglesias examining 828 euthyroid patients with T2DM did not find any relationship between TSH values and metformin treatment in euthyroid T2DM patients [28]. 

Conversely, Cappelli and colleagues have also shown a significant reduction in TSH in euthyroid patients with higher baseline TSH levels that is independent from the presence of AbTPO [14]. 

Studies on the association between metformin treatment and TSH serum levels were also extended to overweight women with polycystic ovarian syndrome (PCOS) and hypothyroidism, and those with chronic Hashimoto thyroiditis. 

Rotondi and colleagues showed that after four months of metformin administration, only PCOS women with overt or subclinical hypothyroidism showed a significant reduction in TSH, independent from thyroxine treatment, while the serum levels of TSH and FT4 did not change significantly in euthyroid patients with PCOS after treatment with metformin [11]. Consistent with other previous studies, free TH concentrations and BMI were steady in all three groups. Similarly, Morteza and colleagues showed that TSH levels markedly fell in PCOS women compared to the placebo group after 6 months of metformin treatment, while free TH concentrations were similar to baseline [12]. 

Another study enrolled 255 newly diagnosed T2DM drug-naïve patients and 170 patients with normal thyroid function, 85 patients with hypothyroidism on thyroxine therapy for more than 6 months, and 80 euthyroid T2DM patients receiving metformin for more than 5 years. The results showed that metformin lowered TSH only in patients with preexisting thyroid dysfunction. The administration of metformin did not lead to any significant fall in TSH level, except in AbTPO positive patients, who showed a statistically significant decrease in TSH level after six months of treatment with metformin. These data suggested that the suppressive effect of metformin emerged only in patients with preexisting thyroid dysfunction, which appears to include those with chronic autoimmune thyroiditis [29].

The strengths of the listed works converge on the fact that treatment with metformin determines a reduction in TSH values in the different populations examined: the data already present in the literature concerns hypothyroid patients; our study instead confirms that this reduction effect occurs also in the euthyroid population.

Our study shows a correlation among thyroid volume and HOMA, BMI, WC, TSH and FT4 less than 50%, despite its significance. This weak correlation may be due to the fact that there are numerous variables capable of influencing the parameters we observed in this study. Therefore, despite the statistical significance, the weak correlation indicates that this aspect of our study could be negligible.

Furthermore, our study demonstrates that treatment with metformin can improve thyroid morphology by causing a reduction in thyroid and thyroid nodule volumes. This effect finds its rationale in the concomitant decrease in TSH serum levels found in the patients enrolled, which reduces the proliferative and mitotic stimulus on healthy thyroid cells and those of the nodule.

Data from the literature suggest that metformin can also ameliorate the morphological abnormalities of the thyroid by improving insulin sensitivity [17,27,28]. Recently, in a cross-sectional study, a relationship between glycated hemoglobin and thyroid volume was discussed [30], supporting the idea that metformin may play a role in the adjuvant therapy of proliferative disease. A retrospective study on 66 women with insulin resistance and nodular hyperplasia [22] showed a higher reduction in nodule size in patients who were treated both with metformin and thyroxine compared to patients treated with metformin alone. Furthermore, in the same population, along with the shrinking effect of metformin on thyroid nodules, TSH was found lower and HOMA normalized. The administration of metformin to patients with obesity and insulin-resistance treated with thyroxine for diffuse/nodular goiter provided a significant reduction in TSH and metabolic parameters when compared to patients receiving thyroxine alone, without appreciable differences in thyroid morphology after 6 months [31]. In the study by Rezzónico et al. [22] only women with insulin resistance and thyroid nodules were enrolled. Oleandri et al. [21] involved a short group of 28 patients with abdominal obesity who were treated with metformin for three months. 

Several hypotheses have been advanced to explain the molecular mechanism underlying the effect of metformin on the morphological abnormalities of the thyroid. Some authors reported that insulin-like growth factor (IGF1) is actively involved in the TSH-mediated proliferation of thyrocytes [32]. The insulin/IGF1 signaling pathway is known to modulate the regulation of thyroid gene expression and might additionally act on thyrocyte proliferation and differentiation [33]. Metformin likely exerts a direct antiproliferative effect on the thyroid through the suppression of mTOR activity [34]. Moreover, other evidence suggests that metformin treatment reducing TSH levels has a protective effect on thyroid enlargement, above all in patients with higher levels of urinary iodine concentrations [35].

Thus, given the role of TSH and insulin resistance in nodule formation, metformin may be also considered an effective drug for the prevention or treatment of thyroid nodular disease.

Several hypotheses have been proposed to explain the association between metformin and the decline in TSH levels. However, to date, a common theory has not yet been provided. Among the proposed mechanisms, metformin might change the affinity or the expression of TH receptors, increase the central dopaminergic tone [36], or induce activation of the TSH receptor, hence, increasing the effects of TH on the pituitary [37]. Furthermore, evidence shows that metformin, given its ability to cross the blood–brain barrier and reach the pituitary at a high concentration, can augment TH action centrally. However, a peripheral effect cannot be excluded [38]. Of note, the case report of Krysiak and Okopien [37] shows that the use of metformin in a patient with diabetes and generalized resistance to TH was associated with a significant reduction in TSH and TH levels. Furthermore, this enhanced sensitivity to TH effects resulted in an increased heart rate and basal metabolic rate, as well as sex hormone-binding globulin levels. Another possible explanation of the TSH-lowering effect of metformin may be related to a metformin-induced activation of the adenosine monophosphate-activated protein kinase (AMPK), which participates in several cellular functions and regulates cellular energy metabolism [39]. Furthermore, it is unlikely that the AMPK system is involved in the central effects of metformin on the TRH/TSH system. Metformin exerts an inhibitory effect on AMPK activity in the hypothalamus, where it opposes T3 [40]. Additionally, evidence demonstrates that insulin resistance can have a role in the link between metformin and TSH. In a four-month pilot study, metformin treatment led to a reduction in TSH in patients with interferon-induced hypothyroidism, more pronounced than in patients with Hashimoto’s thyroiditis [41]. Nevertheless, its clinical significance is still unknown. As aforementioned, metformin may modulate thyroid function at the central level. A prospective study showed that metformin was clinically effective in patients with T2DM and untreated amiodarone-induced hypothyroidism and a poor tolerance of exogenous thyroxine [42]. However, some recent evidence stated that the relationship between TSH values and metformin administration may not be independent, and more targeted studies are needed to explain the mechanisms of action of the drug in cases with an activated TSH axis. Given the impact of metformin on the improvement of insulin sensitivity, evidence indicates that metformin also has a beneficial effect in the prevention and management of cancer. Clinical trials in patients with diabetes and thyroid cancer have shown that metformin treatment results in a higher remission rate and survival [43]. Furthermore, in patients with diabetes and cervical lymph node metastasis of differentiated thyroid cancer, metformin was proven to have positive effects [44]. Metformin showed an antiproliferative effect in primary thyrocytes and thyroid cancer cells by reducing hyperinsulinemia and by the inhibition of cell cycle progression and induction of apoptosis [45]. 

Taken together, these data suggest that metformin treatment can be an effective drug, not only for the control of T2DM and/or insulin resistance, with or without proliferative thyroid disease, but also for patients with metabolic syndrome and obesity. 

In conclusion, our study is among the first in the scientific literature to show that metformin treatment can significantly reduce TSH and increase FT4 and FT3 serum levels in patients with normal thyroid function. Metformin also appeared to reduce the proliferative and mitotic stimulus on thyrocytes. Therefore, metformin treatment can determine not only a serum TSH reduction, but also a thyroid and nodule volume decrease. However, further well-designed prospective studies on a higher number of patients that include mechanistic evaluation are needed to confirm these findings.

## Figures and Tables

**Figure 1 pharmaceutics-14-01987-f001:**
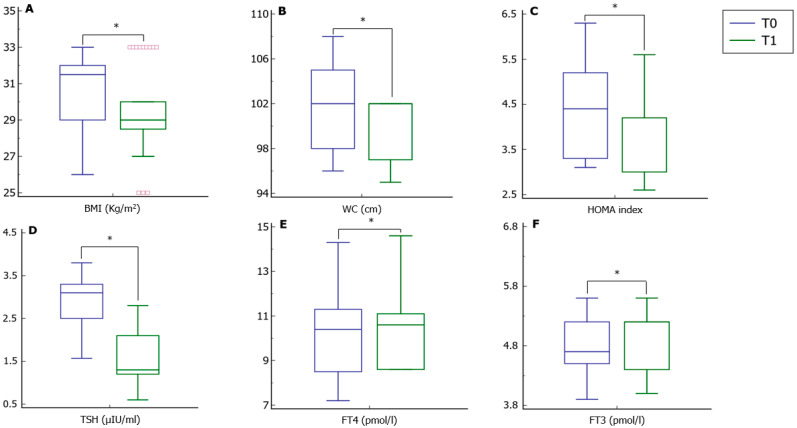
Anthropometric parameters and thyroid function tests before and after treatment with metformin. Panels (**A**–**C**) show a significant reduction in the body mass index (BMI), waist circumference (WC), and homeostatic model assessment (HOMA) index after treatment with metformin, respectively. Panels (**D**–**F**) show a significant reduction in the thyroid-stimulating hormone (TSH) and a significant increase in both free-thyroxine (FT4) and free-triiodothyronine (FT3) serum levels after treatment with metformin, respectively. T0, values at baseline; T1, values after six months. * *p* < 0.05.

**Figure 2 pharmaceutics-14-01987-f002:**
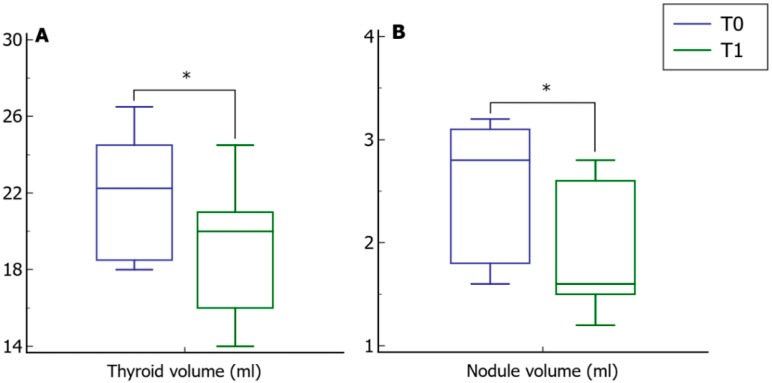
Thyroid and nodule volumes before and after treatment with metformin. Panels (**A**,**B**) show a significant reduction in the thyroid nodule volume following treatment with metformin, respectively. T0, values at baseline; T1, values after six months. * *p* < 0.05.

**Figure 3 pharmaceutics-14-01987-f003:**
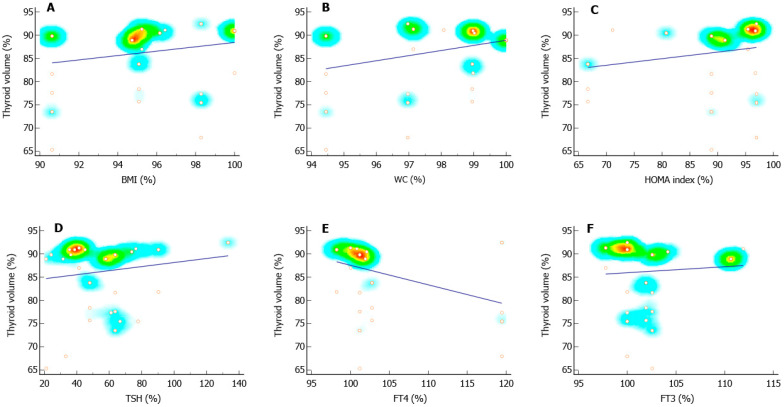
Scatter plots and regression lines of the thyroid volume ratio. Panels (**A**–**C**) show the distribution and regression line of the thyroid volume ratio (%) and the ratio of the body mass index (BMI) (**A**), the waist circumference (**B**), and the homeostatic model assessment (HOMA) index (**C**). Panels **(D**–**F**) show the distribution and regression line of thyroid volume ratio (%) and the ratio of thyroid-stimulating hormone (TSH) (**D**), thyroxine (FT4) (**E**), and free-triiodothyronine (FT3) (**F**). The ratio of each variable was calculated using the following formula: [(after value/before value) × 100].

**Figure 4 pharmaceutics-14-01987-f004:**
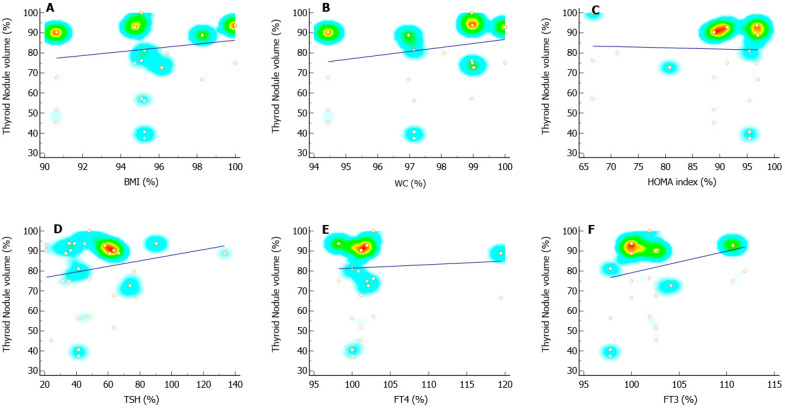
Scatter plots and regression lines of the ratio of the nodule volume. Panels (**A**–**C**) indicate the values distribution and regression line of the ratio of the thyroid nodule volume (%) and the ratio of the body mass index (BMI) (**A**), the waist circumference (**B**), and the homeostatic model assessment (HOMA) index (**C**). Panels (**D**–**F**) indicate the values distribution and regression line of the ratio of the thyroid nodule volume (%) and the ratio of thyroid-stimulating hormone (TSH) (**D**), thyroxin (FT4) (**E**), and free-triiodothyronine (FT3) (**F**). The ratio of each variable was calculated using the following formula: [(after value/before value) × 100].

**Table 1 pharmaceutics-14-01987-t001:** Baseline parameters of the enrolled cohort.

	Overall Cohort	Male	Female
*n*	50	10	40
Age (years)	36.9 ± 12.8	30.5 ± 12.1	38.5 ± 12.6
BMI (Kg/m^2^)	30.6 ± 2.0	30.3 ± 1.8	30.7 ± 2.1
WC (cm)	102.1 ± 4.1	105.0 ± 3.2	101.4 ± 4.1
HOMA index	4.5 ± 1.2	5.5 ± 0.9	4.2 ± 1.1
TSH (µIU/mL)	2.9 ± 0.6	3.6 ± 0.3	2.8 ± 0.6
FT4 (pmol/L)	10.1 ± 1.8	9.5 ± 1.0	10.2 ± 1.9
FT3 (pmol/L)	4.7 ± 0.6	4.3 ± 0.4	4.9 ± 0.6
Thyroid volume (mL)	22.2 ± 2.8	21.3 ± 3.4	22.4 ± 2.6
Nodule volume (mL)	2.5 ± 0.6	3.0 ± 0.6	2.3 ± 0.6

Data are reported as mean ± SD; BMI, body mass index; WC, waist circumference; HOMA, homeostatic model assessment; TSH, thyroid stimulating hormone; FT4, thyroxin; FT3, free-triiodothyronine.

**Table 2 pharmaceutics-14-01987-t002:** Multiregression analysis of thyroid volume.

	Independent Variables	Coeff.	Std. Error	t	*p*	r_partial_	r_semipartial_	VIF
	(Constant)	14.7623						
Ratio (%)	BMI	−0.5055	0.6815	−0.742	0.4623	−0.1124	0.08816	7.013
	WC	1.7123	1.0770	1.590	0.1192	0.2356	0.1890	6.757
	HOMA index	0.2338	0.1068	2.188	**0.0341**	0.3165	0.2601	1.476
	TSH	0.1062	0.04047	2.625	**0.0119**	0.3716	0.3120	1.365
	FT4	−0.5227	0.1531	−3.414	**0.0014**	−0.4618	0.4057	1.382
	FT3	−0.2007	0.3421	−0.587	0.5604	−0.08913	0.06974	3.114
	(Constant)	0.6420						
Difference (Δ)	BMI	−0.8474	0.6184	−1.370	0.1777	−0.2045	0.1569	9.362
	WC	0.6245	0.2729	2.288	**0.0271**	0.3295	0.2621	8.017
	HOMA index	1.1012	0.7080	1.555	0.1272	0.2308	0.1782	1.939
	TSH	0.7785	0.3806	2.045	**0.0470**	0.2978	0.2343	1.880
	FT4	−1.9855	0.5246	−3.785	**0.0005**	−0.4999	0.4335	1.373
	FT3	−0.5183	1.7888	−0.290	0.7734	−0.04415	0.03319	2.860

**Table 3 pharmaceutics-14-01987-t003:** Multiregression analysis of thyroid nodule volume.

	Independent Variables	Coeff.	Std. Error	t	*p*	r_partial_	r_semipartial_	VIF
	(Constant)	−96.9390						
Ratio (%)	BMI	2.9434	1.8960	1.552	0.1279	0.2304	0.2135	7.013
	WC	−2.9136	2.9963	−0.972	0.3363	−0.1467	0.1337	6.757
	HOMA index	−0.1986	0.2972	−0.668	0.5075	−0.1014	0.09191	1.476
	TSH	0.07272	0.1126	0.646	0.5218	0.09802	0.08882	1.365
	FT4	−0.1454	0.4260	−0.341	0.7345	−0.05198	0.04693	1.382
	FT3	2.0563	0.9516	2.161	**0.0363**	0.3130	0.2972	3.114
	(Constant)	0.4233						
Difference (Δ)	BMI	0.4208	0.2032	2.070	**0.0445**	0.3011	0.2698	9.362
	WC	−0.2482	0.2327	−1.067	0.2921	−0.1606	0.1390	1.939
	HOMA index	−0.1214	0.08968	−1.354	0.1828	−0.2022	0.1765	8.017
	TSH	0.05472	0.1251	0.437	0.6640	0.06656	0.05702	1.880
	FT4	−0.02535	0.1724	−0.147	0.8838	−0.02242	0.01917	1.373
	FT3	1.5362	0.5879	2.613	**0.0123**	0.3702	0.3406	2.860

## Data Availability

Data are available upon request to the corresponding author.

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
