# Peer review of "Effects of Slow-Acting Metformin Treatment on the Hormonal and Morphological Thyroid Profile in Patients with Insulin Resistance"

_pharmaceutics, 2022, doi:10.3390/pharmaceutics14101987_

Round 1

Reviewer 1 Report

1)    In the title and along the manuscript, do the authors refers to “extended-release metformin”?

2)    In the abstract, the authors state that metformin is an effective drug for treatment of obesity. This is not adequate, due its weight reduction effectiveness is lesser than 5%.

3)    I suggest replacing the terms “obese” or “diabetic”. Use instead: “patients with obesity” or “patients with diabetes”

4)    Did the authors evaluate the differences among males and females in parameters of table 1? Please, add p value

5)    Please add the p value in each panel of Figure 1. In its current state, it seems to be no differences in FT4 and FT3. Also add p value for Figure 2

6)     According to the table 1, the correlation among thyroid volume and HOMA, WC, TSH and FT4 are lesser than the 50% (despite its significance). These results should be proper discussed into the proper section. The same for thyroid nodule volume and FT3, BMI and FT3

7)    In the discussion, the authors state that metformin cures type 2 diabetes. This is overestimated, it controls it. Furthermore, I suggest decreasing the comparative with previous literature and highlighting their results, also adding the strengths and limitations of their study.

Author Response

Please, see attached file.

Reviewer 2 Report

Manuscript #762218 concerns the impact of metformin intervention on thyroid gland function. I am not sure if the topic is suitable to the journal, maybe other journal covering clinical research could be suitable. Nevertheless, several parts of the manuscript require revision in order to add clarity or to make the manuscript better focused. I have listed some concerns/suggestions below:

1.     The study involved insulin resistant patients – this should be clearly stated in the manuscript title.

2.     The introduction section is too long, contains the background that suggests this study is a repetition. Metformin characteristics, its pleiotropic action, any data gaps, clearly stated aim of the study should be included in the introduction section. 

3.     Additional references should be given in the introduction section i.e. ‘Metformin appears to reduce TSH levels in untreated hypothyroid patients. This aspect could be related to metformin-induced weight reduction.’.

4.     ‘To date, there is a lack of data in the scientific literature on the possible effect of metformin on thyroid volume and nodules on a euthyroid population.’. This is not true - in previous sentence the authors cite the manuscript ‘Metformin Decreases Thyroid Volume and Nodule Size in Subjects with Insulin Resistance: A Preliminary Study’. 

5.     Manuscripts 10.3390/cancers14051336, 10.3389/fendo.2019.00465, 10.1007/s40618-019-01059-w,10.2174/1381612825666190918162649 should be included in the bibliography/discussed.

6.     Moreover, hypocaloric diet, weight loss, and lifestyle changes are also well-known factors that affect thyroid nodules. How can authors confirm that the effects observed were the separate effects of metformin treatment not the diet/glycemia? Additional control group should be included. In my opinion, no conclusions can be drawn without the control group.

7.     Results section – T0 and T1 should be described in the text.

8.     Fig 1A – what do the pink squares mean in the T1 BMI box plot?

9.     Why did the authors introduce the Δ values and the ratios? Multiregression analysis did not show that the thyroid volume is not associated with the diet of BMI. It only shows that the thyroid volume is or is not associated with the ratios/Δ of the variables. Additional multiregression analysis for variables (not ratios/ Δ) should be included.

10.  Discussion section needs to be improved. It looks like several paragraphs containing some sentences (or just one sentence). It is necessary to relate these sentences, to discuss the results obtained and the results already published, to discuss the hypothesis that was drawn in the introduction, to describe any discrepancies between the studies, to point the strengths and weaknesses of the studies, and to draw some conclusions.

11.  Extensive english editing is needed.

Author Response

Please, see attached file.

Round 2

Reviewer 2 Report

Thank you for your comprehensive answers to my questions. I appreciate the contribution of the work to the manuscript. The authors addressed the suggestions to the previous version of the manuscript and in my opinion the quality of the manuscript has improved significantly.